# Large-Scale LiDAR SLAM with Factor Graph Optimization on High-Level Geometric Features

**DOI:** 10.3390/s21103445

**Published:** 2021-05-15

**Authors:** Krzysztof Ćwian, Michał R. Nowicki, Jan Wietrzykowski, Piotr Skrzypczyński

**Affiliations:** Institute of Robotics and Machine Intelligence, Poznan University of Technology, 60-965 Poznan, Poland; michal.nowicki@put.poznan.pl (M.R.N.); jan.wietrzykowski@put.poznan.pl (J.W.); piotr.skrzypczynski@put.poznan.pl (P.S.)

**Keywords:** 3-D LiDAR, SLAM, geometric features, optimization, bundle adjustment

## Abstract

Although visual SLAM (simultaneous localization and mapping) methods obtain very accurate results using optimization of residual errors defined with respect to the matching features, the SLAM systems based on 3-D laser (LiDAR) data commonly employ variants of the iterative closest points algorithm and raw point clouds as the map representation. However, it is possible to extract from point clouds features that are more spatially extended and more meaningful than points: line segments and/or planar patches. In particular, such features provide a natural way to represent human-made environments, such as urban and mixed indoor/outdoor scenes. In this paper, we perform an analysis of the advantages of a LiDAR-based SLAM that employs high-level geometric features in large-scale urban environments. We present a new approach to the LiDAR SLAM that uses planar patches and line segments for map representation and employs factor graph optimization typical to state-of-the-art visual SLAM for the final map and trajectory optimization. The new map structure and matching of features make it possible to implement in our system an efficient loop closure method, which exploits learned descriptors for place recognition and factor graph for optimization. With these improvements, the overall software structure is based on the proven LOAM concept to ensure real-time operation. A series of experiments were performed to compare the proposed solution to the open-source LOAM, considering different approaches to loop closure computation. The results are compared using standard metrics of trajectory accuracy, focusing on the final quality of the estimated trajectory and the consistency of the environment map. With some well-discussed reservations, our results demonstrate the gains due to using the high-level features in the full-optimization approach in the large-scale LiDAR SLAM.

## 1. Introduction

The knowledge of the robot pose and the availability of an environment model are enabling conditions for efficient task and motion planning, reasoning about the environment’s semantics, and human–machine interaction in autonomous robot vehicles. Considering a previously unknown environment, the robot has to solve the simultaneous localization and mapping (SLAM) problem. There is a large body of research on SLAM and related problems (e.g., visual odometry). Some simple variants of SLAM, e.g., in 2-D structured environments, are considered solved, however, there are many open issues when it comes to more complicated yet more practical cases, including 3-D scenes, dynamic environments, and life-long map learning [1].

In particular, urban environments with high buildings, tunnels, and underground parking lots require localization solutions that are independent of the global positioning system (GPS) signals. SLAM systems for autonomous vehicles, such as self-driving cars or delivery robots, should yield accurate estimates of the trajectory in 3-D, outdoors, and indoors, employing the limited onboard sensors of the vehicle [2].

Whereas a significant part of the SLAM research in the last two decades was devoted to passive visual perception [1], the active 3-D laser sensors (known also as light detection and ranging—LiDAR) have important practical advantages compared to the passive cameras. The active sensors are well-suited to work under any lighting conditions, while passive cameras struggle with poorly illuminated scenes and sudden lighting changes. Although passive cameras are certainly the most affordable and easy to integrate sensors, LiDARs are considered an enabling technology in the growing self-driving vehicles sector [3], and can be indispensable in other practical applications of robotic platforms, for example, underground inspections [4].

An important issue in LiDAR SLAM is the environment map representation. Currently, LiDAR SLAM solutions either employ raw point clouds or extract some geometric features from these clouds. Salient point features are widely employed in visual SLAM. Such systems, such as ORB-SLAM2 [5], detect salient points in images, match these features to the map, and then further optimize their position, which results in accurate estimates of the camera pose. Unfortunately, extracting salient point features from point clouds is computationally intensive and prone to errors due to the uncertainty in measured range [6]. However, it is possible to efficiently extract from the LiDAR point clouds some higher-level geometric features, namely line segments and planar patches.

These spatially extended features group many measured points and are more robust to the range measurement errors, because points that do not match any feature can be rejected at an early processing stage. A SLAM system employing higher-level features should be also characterized by increased accuracy, because spatially extended features provide better geometric constraints than point-to-point correspondences characteristic to the methods processing raw point clouds.

Moreover, the map representation with geometric features is relevant to the optimization frameworks, which can be applied for the refinement of both the robot trajectory estimate and the feature estimates in the map at the same time. In LiDAR SLAM non-linear optimization is often used to register together the subsequent laser scans iteratively, or to register the incoming scans to a static map. Point clouds obtained from modern LiDARs can contain tens of millions of points per second, and therefore it is impossible to use these raw points as nodes of a factor graph that is optimized, such as the point features in visual SLAM, employing bundle adjustment, and optimizing jointly the robot trajectory and the map. On the other hand, the bundle adjustment approach improves the accuracy of the feature-based visual SLAM solutions [7]. This can be an inspiration for LiDAR SLAM, but an appropriate map representation is required.

Therefore, we developed PlaneLOAM, a LiDAR SLAM system that uses higher-level features that group the raw points and describe them as simple geometric objects. The map in our system consists of planar patches and line segments that can be related to the robot trajectory by observations and make it possible to build a factor graph with constraints due to the sensor motion and observations of features. In order to exploit re-observations of the already visited parts of the environment, we implemented a loop detection mechanism, which is based on SegMap [8]. This approach learns robust descriptors of the segmented point clouds, which are then used to determine the re-observed locations in the map. The additional observations that join the previously unrelated features introduce important new constraints to the factor graph, and allow our system to consistently map large environments.

Although we propose new features, a new map structure, and its optimization procedure, the overall software architecture of PlaneLOAM retains a similarity to the LiDAR odometry and mapping (LOAM) system [9], which combines real-time scan-to-scan pose tracking (i.e., odometry) and slower, but more accurate scan-to-map localization. In our recent conference paper [10] we have demonstrated the feasibility of the new map representation, and we evaluated the improvement in trajectory estimation accuracy with respect to the LOAM system. This article introduces PlaneLOAM 2.0, a version extended with the detection of loop closures and map optimization employing high-level features. We also evaluate the new version in a much broader application context. The proposed solution is evaluated on three different datasets employing different LiDAR sensors. We compared it against the open-source LOAM system and tested three versions of PlaneLOAM 2.0: without loop closures, following the LOAM approach but with higher-level features, and with loop closures implemented either using a simple pose graph or employing the factor graph approach. The contribution of this article can be summarized as:robust data association methods for creation of and updating the planar and linear features,efficient map management procedures that make it possible to build large, feature-based maps,loop closing using robust descriptors for loop detection and geometric features for map optimization,thorough evaluation of the proposed solution in different variants and in several representative scenarios.

The remainder of this paper is organized as follows: Section 2 reviews the most relevant related work in LiDAR SLAM, Section 3 introduces the architecture of the proposed SLAM system, while Section 4 describes the loop closing mechanism. Then, Section 5 presents the experimental evaluation of our system on three different data sets with different LiDARs, and Section 6 concludes the paper highlighting the gains due to the feature-based representation.

## 2. Related Work

Since it is not possible to review here the overwhelming amount of work in SLAM published in the last two decades, we focus on the prior works that are directly related to our research in three aspects: the general architecture and map representation, the methods used for registration of consecutive scans, and the approaches to detect and close the loops whenever the robot revisits an already mapped area. To put our research in a proper application context, we focus on methods and systems solely based on LiDAR data, and applied to ground robots or autonomous vehicles. Those interested in a more general survey should consider the work of Cadena et al. [1], which is however mostly focused on visual SLAM, and our earlier survey paper [11].

### 2.1. Architectures of LiDAR-Based SLAM

In the state-of-the-art visual SLAM efficient management of the historical data in the environment map is crucial to the achieved accuracy [12], because optimization is preferred to filtering (either a variant of Kalman filter or particle filter). Joint optimization of the sensor (robot) trajectory and the map makes it possible to update the linearization points and alleviate the negative effects of approximations and incorrect measurements by employing robust cost functions [13]. In the contrary, most of the LiDAR-based SLAM systems employ raw point clouds to represent the map. This is often a practical choice, as the raw-point-based representation does not rely on the existence of any specific features in the environment, and when combined with loop closure detection and global, pose-based optimization results in very accurate maps [14]. However, approaches like [14] can run in real-time only if massively parallel processing on a general purpose graphic processing unit (GPGPU) is employed [15]. Moreover, raw point-based representations cannot distinguish salient features, such as objects of defined classes or distinct geometric primitives, which in turn makes it hard to track those features over multiple scans in a similar way to visual tracking, and eventually reduces the number of constraints that can be established in the map and then used by the optimization. Several prior works attempted to use volumetric or surfel-based map representations with LiDAR data. A voxel grid-based representation with subsampled point clouds was used in [16]. The normal distribution transform (NDT) was also used for compact 3D model (map) representation and scan-to-model registration [17]. A drawback of the grid-based map representations is that global nearest neighbors search has to be used to establish correspondences between the scan points and the model, which renders these approaches computationally inefficient. Map representation with surface elements (surfels) in SLAM was popularized by the successful RGB-D-based ElasticFusion method [18], which has directly inspired work on LiDAR-based systems [19]. Droeschel and Behnke [20] hybridized the surfel representation with a local, robot-centric multi-resolution grid map. The map has higher resolution in the proximity to the LiDAR, which makes it possible to efficiently aggregate the scanned points to surfels, and then to use the local maps in a multi-level graph-based SLAM. The surfel-based representation allowed Behley and Stachniss [21] to exploit the projective data association between the laser scan and a rendered view from the map, which enables real-time mapping at a large scale.

Volumetric representations are also used in recent attempts to solve the LiDAR-based SLAM or LiDAR-based odometry problems employing deep learning methods. Unfortunately, standard representations of the LiDAR data, such as point clouds, are not differentiable [22]. Hence, state-of-the-art neural models used for the processing of range data ensure invariance to the point-order permutation, like PointNet [23], or employ a voxel representation [24]. While the PointNet approach and its derivatives make it possible to process point clouds with convolutional networks, they are considered as taking too much of the computing resources to be applied in real-time SLAM. Among the few attempts to solve LiDAR-based localization in an unknown environment (either odometry or SLAM) with neural network architecture, Velas et al. [25] transformed the pose estimation problem into a classification task, while the recent L${}^{3}$-Net system [26] combines a lightweight variant of the PointNet with another neural network that accumulates the matching costs from all the features in a volumetric model and outputs the pose. Unfortunately, casting the pose estimation problem as classification or cost accumulation leads to decreased accuracy of the pose estimate, because the result is bounded to discrete values depending on the data structure being used. The recent DeepLO method [27] offers pose estimation accuracy comparable to model-based systems, and uses an unsupervised loss function based on the iterative closest points formulation, leveraging also the normal vectors consistency to obtain a confidence map that provides weighting factors to the loss function. While unsupervised learning marks important progress in learnable LiDAR SLAM, the DeepLO had to be trained separately for the three different data sets it was tested on, which in turn points to a problem with entirely learnable LiDAR-based SLAM: it may experience problems when confronted with a previously unseen environment.

In most of the contemporary LiDAR-based odometry or SLAM systems, the map is not optimized at all or is optimized only as a graph of the previous sensor/robot states (poses). The raw points or volumetric representation does not facilitate global optimization with the marginalization of the historical states, which keeps the size of the map bounded, and efficient management of the historical data in the map, which is crucial to the achieved accuracy in visual SLAM based on bundle adjustment (BA) optimization [28]. In visual SLAM the map is either organized as a collection of images (direct approach), or made up of sparse feature points in a graph structure [12]. While there exist detectors of 3-D salient point features that can be applied to point clouds [6], they are too computationally intensive to be used for large scenes in outdoor SLAM.

Few LiDAR-based SLAM systems attempt to structure the map beyond points. The LiDAR odometry and mapping (LOAM) [9], one of the most influential architectures in LiDAR-based localization, segments the acquired scans into semantically distinct planar patches and lines, and then applies different metrics to these classes in the iterative closest points registration process. LOAM does not create geometric features in the map, which is represented as a large point cloud over a regular grid for improved search in data association. The innovation of LOAM is also the software architecture that combines real-time scan-to-scan sensor pose tracking (incremental odometry) and slower, but more accurate scan-to-map localization. The careful choice of points that represent well-defined, static parts of the environment and the ability to improve the incremental odometry by model-based registration make LOAM one of the most accurate real-time LiDAR-based localization systems. Therefore, LOAM inspired many researchers to improve some aspects of the system retaining the overall architecture with the separated odometry and mapping threads, including our PlaneLOAM system introduced in [10] and extended in this paper. Among other developments, LeGO-LOAM [29] explicitly detects the ground plane and distinguishes between points measured on the ground and on other objects. Moreover, knowing the ground plane, it determines the sensor attitude (pitch and roll) with respect to this plane, and then estimates the remaining components of the six degrees of freedom (d.o.f) sensor pose from a simplified formulation. LLOAM [30] adds to the LOAM architecture a loop closing module that is based on point clouds segments with learned descriptors [31], which is similar to our approach, but LLOAM builds only a pose-graph based on the sensor poses and relative transformations calculated by the LOAM-like odometry, while we exploit the geometric features to construct a factor graph, like in BA-based visual SLAM. The system described in [32] also combines LOAM-inspired LiDAR odometry and mapping with the SegMatch [31] approach to loop detection. It adds some ground plane constraints to improve the estimation of the relative pose transformations, but still uses a simple pose graph structure to handle the detected loop closures.

A natural approach for LiDAR data in partially structured urban environments are high-level geometric features, such as planar and linear segments. Planar features in EKF-based SLAM with 3-D laser data were used by Weingarten and Siegwart [33]. Later the use of planar segments in a pose-graph SLAM framework was demonstrated in [34], leveraging a fast method for registration of noisy planar features with unknown correspondences [35]. Grant et al. [36] also used a fast plane detector to extract high-level features from LiDAR data. This SLAM method uses the extracted planar features as landmarks in a factor graph optimization problem, adding also laser odometry factors between the sensor poses in order to make the system capable of working in less structured environments. Loops are detected by determining the similarity of the planar features stored in a global map, and then they are closed by establishing constraints (factors) between similar features and optimizing the graph. The approach described in [36] is most similar to PlaneLOAM 2.0, however it differs with regard to a number of the architecture components. For the sake of computational efficiency, PlaneLOAM keeps the general LOAM processing scheme with separated threads for scan-to-scan and scan-to-map matching, with map optimization as an additional process. Our map contains both planar and linear segments (like in LOAM, but at a higher abstraction level), and we use highly descriptive point cloud segments for loop detection. The latter feature allows PlaneLOAM to close large and multiple loops, which is rather impossible with the approach from [36], because determining the similarity of planar features upon a number of geometric tests is more prone to errors caused by trajectory drift, changed viewpoint, and occlusions than matching of learned descriptors.

### 2.2. Scan Registration Methods

The most popular approaches to estimate the relative transformation between two scans are variants of the iterative closest points (ICP) method [37]. Despite its wide use and many improvements [38], the ICP has high computational cost and is prone to data association errors because of the need to re-establish the associations by search once the sensor pose is re-estimated. This problem can be alleviated by carefully choosing the points for matching. In this context, LOAM applies the point-to-plane and point-to-line distance metrics to points described as belonging to planes and lines, respectively [9], while IMLS-SLAM [39] uses implicit moving least squares surfaces and matches points selected by a criterion that exploits the normal vectors to constrain all six degrees of freedom of the sensor. It is shown in [39] that keeping too many points that do not provide useful constraints (e.g., because they belong to non-stationary objects or have large measurement errors) degrades the accuracy of the computed transformation. We consider this in PlaneLOAM, and propose a map representation that facilitates accurate data association by distinguishing between separated objects in structured scenes, which helps to eliminate spurious LiDAR readouts, e.g., those originating from vegetation or dynamic objects.

Another issue that has to be considered while registering consecutive LiDAR scans is handling range readouts that are subject to rolling-shutter-like distortions whenever the scans are taken in motion, which is typical for applications in autonomous vehicles. Hence, either the registration method has to be adopted [40], or the scans need to be undistorted using a motion prior. Some LiDAR SLAM systems exploit tightly integrated IMU (inertial measurements unit) measurements to compensate these distortions [41], but if no data providing external motion estimates are available, the distortions have to be corrected while estimating the sensor motion. This is accomplished by exploring the matching of the consecutive laser scans.

This problem has been addressed in [20] by representing the sensor trajectory as a continuous B-spline in SE(3) and interpolating between the sensor poses. The LOAM and LeGO-LOAM systems take a simpler approach, by assuming a constant velocity of the sensor, which makes the interpolation much easier [9,29]. As this approach is sufficient for a mostly planar motion of ground vehicles, we follow this idea in PlaneLOAM and estimate the sensor motion between two consecutive poses in the LiDAR odometry thread.

### 2.3. Closing Loops in LiDAR SLAM

Although LiDAR-based localization and mapping are researched very actively, relatively few of the published systems can be considered full solutions to the SLAM problem, as they lack the ability to close loops, that is to identify the environment areas already visited by the robot and included in the map. Many of these systems are LiDAR odometry solutions with scan-to-model registration [9,29,39,41], but while they optimize the sensor trajectory and extend the map, the map itself cannot be optimized when canceling the trajectory drift. This is a major limitation in the context of mapping structured urban areas, where the lack of map correction can lead to inconsistencies in the representation of objects.

While using ICP registration between new scans and the map for loop closing is possible [42], this approach is inefficient for closing large loops due to the computation burden of the nearest neighbor search. Appearance-based loop closure detection is more efficient, once a proper data representation of the LiDAR data is established. Magnusson et al. [43] pioneered this approach applying NDT and describing locations with histograms. Droeschel and Behnke [20] detected loops by matching local surfel-based maps, while [21], which also used surfel-based map representation, created virtual views of the global map, enabling projective associations and thus robust detection of re-visited areas even for partial overlapping.

Recently, learned neural models demonstrated their effectiveness in solving the loop closing problem for LiDAR data. The SegMap system [8] represents the laser-scanned environment in the form of point cloud segments with attached learned descriptors. SegMap identifies previously visited places matching a local map of segments obtained from the recent LiDAR scans to a global map of segments. The segments are incrementally grown from the LiDAR points accumulated in a dynamic voxel grid. A deep neural network is used to generate the descriptors of segments [31]. The learned representation used in SegMap is highly descriptive, which makes it possible to match local and global representations unambiguously even for large trajectory drifts. This approach was already applied to LOAM-inspired SLAM architectures [30,32], but only using pose-graph optimization to integrate the sensor pose constraints imposed by the detected loops. In contrast, we are the first to demonstrate the potential of incorporating the SegMap robust loop detection into LiDAR SLAM based on a full, BA-like factor graph optimization. Our PlaneLOAM takes advantage of the descriptive power of the SegMap descriptors to detect the loops, but then leverages the accuracy of the feature-to-feature data registration in order to improve the map and trajectory accuracy beyond the results possible with pose-graph optimization.

## 3. The Architecture of PlaneLOAM

The proposed PlaneLOAM system adapts the similar processing steps to LOAM by dividing the computation into the odometry and mapping module while extending the LOAM’s capabilities by the addition of the loop closing module (Figure 1). We introduce the modules of PlaneLOAM in the following subsections.

### 3.1. Odometry

In the odometry thread, PlaneLOAM processing is the same as for the LOAM algorithm with slight modifications to allow operation for different 3D laser scanners. Firstly, the sets of points belonging either to a plane (planar) or a line (linear) are extracted from the newly captured point cloud. These points are then matched to the closest points of the same type (planar or linear) from the previously captured point cloud. The matching is performed based on the Euclidean distance between points of the same type, accommodating a simple sensor motion prediction model. To achieve real-time performance, the matching is performed with efficient kd-trees to speed up the search process. The matches between points from consecutive point clouds are then used to create constraints for the optimization problem that is solved to determine the pose estimate. Despite the matching being performed on a point-to-point basis, the optimization constraints are based on the point-to-line and point-to-plane distances. As in LOAM, line and plane equations from a previous point cloud are calculated dynamically at each step from the set of five nearest points. The point matching and pose estimation steps are repeated multiple times until the optimization converges.

The optimization problem in the odometry step can therefore be formulated as:(1)$$T^{*}=\underset{T}{\mathrm{argmin}}\left(\sum_{i}f\left(p_{i},t_{i},T\right)+\sum_{j}g\left(p_{j},t_{j},T\right)\right),$$
where $f\left(p_{i},t_{i},T\right)$ represents the point-to-line distance for a point $p_{i}$ with its corresponding acquisition time relative to the beginning of the scan $t_{i}$, $g\left(p_{j},t_{j},T\right)$ stands for the point-to-plane distance for point $p_{j}$ with its corresponding acquisition time relative to the beginning of the scan $t_{j}$, and $T^{*}$ is the best pose estimate.

Unlike global shutter cameras, measurements in LiDARs are done point by point or in a group of points. When the sensor is moving, each measurement in a full scan is performed from a different pose. To facilitate further computations, it is necessary to transform all measurements to one frame of reference by compensating LiDAR’s movement. In this work, we chose to transform points to the LiDAR’s frame of reference at the end of the scan. By assuming that the velocity during the scan was constant, the transformation of a point ${}^{m}p$ expressed in the LiDAR’s frame of reference in the moment of measurement to a point ${}^{e}p$ expressed in the frame of reference at the end of the scan, can be computed as follows:(2)$${}^{e}p=T_{i,i+1}^{-1}\exp\left\{\frac{t_{i}}{t_{s}}\log T_{i,i+1}\right\}{}^{m}p,$$
where $t_{i}$ is a time of the *i*-th measurement relative to the beginning of the scan, $t_{s}$ is a duration of the whole scan, and $T_{i,i+1}$ is a pose at the end of the scan relative to the start of the scan.

The transformation is computed using LOAM’s odometry procedure, $t_{s}$ is equal to 0.1 s for all LiDARs used in this paper and $t_{i}$ depends on the type of sensor used:Velodyne HDL-64E:
(3)$$t_{i}=\frac{\phi_{i}-\phi_{s}}{\phi_{e}-\phi_{s}}t_{s},$$
where $\phi_{i}$ is a horizontal angle of the *i*-th measurement, $\phi_{s}$ is a horizontal angle of the start of the scan, and $\phi_{e}$ is a horizontal angle of the end of the scanSick MRS 6124:
(4)$$t_{i}=\left(\frac{\left\lfloor g/6\right\rfloor }{4}+\frac{\phi_{i}-\phi_{s}}{2\pi }\right)t_{s},$$
where *g* is a number of a scanning ring, counting from the top.Ouster OS1-64:
(5)$$t_{i}=\frac{\left\lfloor i/64\right\rfloor }{1024}t_{s},$$
where *i* is an index of the measurement.

### 3.2. Mapping

The LOAM system divides registered scan points into linear and planar points that are used to perform optimization in both odometry and mapping. However, they do not form higher-level features as they are stored in a form of unordered point clouds for both types. When necessary, the five closest points of a selected type from these aggregated point clouds are used to form a constraint.

The proposed PlaneLOAM adapts an alternative map representation that groups points into high-level features. This approach does not limit point number, which means that a single feature can represent a large object, such as a wall of the building or a road surface. A conceptual comparison between the PlaneLOAM representation of a wall and the representation in the original LOAM is shown in Figure 2.

#### 3.2.1. Map Representation

Map of the environment, created in PlaneLOAM, consists of both *linear* and *planar* features. However, the presented approach affects mostly planes as they are more likely to form large structures regardless of a type of surrounding. A diagram presenting the structure of the stored map is shown in Figure 3.

Each planar feature is initially created using five nearby points that were registered during a single laser scan. Its equation ($n,n_{d})$ is four-dimensional where n is the unit-length plane normal and $n_{d}$ is the distance of a plane from the origin. This equation is called a plane equation for brevity and satisfies the condition:(6)$$p\cdot n+n_{d}=0$$
for all points p lying on a plane. The normal vector n is calculated using PCA (principal component analysis) of a given feature’s covariance matrix and it is equal to the eigenvector corresponding to the smallest eigenvalue. The covariance matrix is calculated with respect to the given feature’s centroid and measures the spread of all points belonging to that feature.

Line segments are represented by six-dimensional Plücker coordinates $\left[\begin{array}{cc} l_{d} & l_{m} \end{array}\right]$, where $l_{d}$ is direction of the line, and $l_{m}$ is the moment of the line (both three-dimensional) with respect to the global frame of reference. The line direction $l_{d}$ is initially created based on two points $p_{1}$, $p_{2}$ using the following equation:(7)$$l_{d}=\frac{p_{2}-p_{1}}{∥p_{2}-p_{1}∥}.$$
The moment of the line $l_{m}$ is calculated as:(8)$$l_{m}=\frac{p_{1}\times p_{2}}{∥p_{2}-p_{1}∥}.$$
Depending on the feature size, $p_{1}$, $p_{2}$ are either two real points or two artificial points chosen to lie on the line direction $l_{d}$ equal to the eigenvector corresponding to the largest eigenvalue from PCA.

Both plane and line parameters are updated every time new points are added to the feature, but only until the number of its points is smaller than 30. After that threshold, the parameters of an equation representing the feature remain constant. Each feature stores additional information, that was also computed during the calculation of the equation, such as its centroid and the covariance matrix of points. Moreover, each feature has also coefficients describing *planarity* or *linearity* (depending on the type of feature), which is a measure of its quality. *Planarity* and *linearity*, denoted by *p*, is the percentage of points that are closer to the feature than a certain value (0.2 m in the case of our system). It is calculated with the use of the following equation:(9)$$p=\frac{m}{N}\cdot 100\%,$$
where *m* is the number of points within a distance of 0.2 m from the estimated line or plane while *N* is the number of points belonging to the selected feature. That coefficient is used to verify whether a given feature is valid and to check how scattered the points are. The value of the parameter *p* and hence the quality of the feature might be improved by deleting points whose distance to the plane or line exceeds a given threshold. The processing of all created features is performed in several sequential steps, consisting of creating, updating, deleting, and merging. This life-cycle of the high-level features is presented in the block diagram shown in Figure 4.

#### 3.2.2. Creating Features

The first step of the processing pipeline is the creation of planar and linear features from the points registered in each scan. To create a plane, we search for five nearby points and calculate the plane equation based on their coordinates. Although it is possible to use only three points to determine a plane equation, a bigger number of points improves accuracy and reduces the chance of forming invalid features. Since the feature has assigned parameters of the planar equation, all newly added points must satisfy the model specified by these parameters with defined tolerance. The process of adding points to features is presented in Figure 5.

Initially, for every new scan point, we calculate the point-to-plane or point-to-line distance to the three nearest features and try to assign that point to one of them. The point-to-plane distance is calculated using following equation:(10)$$d_{1}=f\left(p,T\right)=∥\left(Tp\right)\cdot n+n_{d}∥,$$
where p is a vector of considered point and T is a transformation moving a point p from the local coordinate system to the global coordinate system. In the case of linear features, the point-to-line distance is calculated as
(11)$$d_{1}=g\left(p,T\right)=∥\left(Tp\right)\times l_{d}-l_{m}∥.$$
Based on that distance, the algorithm rejects all features that are further away than 0.6 m and then finds three closest ones. The next step is to find the minimal point-to-point distance between the considered point and points belonging to these features. That distance must be smaller than 0.7 m, which prevents from adding points that coincidentally satisfy the feature’s equation. Finally, the algorithm checks if there is more than one feature that complies with these requirements. In that event, it calculates the ratio of point-to-point distances in order to check if one of them is considerably smaller, according to the equation:(12)$$\frac{d_{2min}}{d_{3min}}>0.7,$$
where: $d_{2min}$ and $d_{3min}$ are the distances between the given point and closest points in the first and second nearest feature, as presented in Figure 6. If the equation above is not satisfied, the considered point is not assigned to any of these features, as incorrect correspondences might worsen the localization estimate. All the above conditions must be met so that a new point could be added to the map. The first condition assures that all points, that belong to a given feature, comply with its equation. The second one prevents from forming features with an inconsistent density of points, while the last one provides confidence that matching was performed correctly.

If the above-mentioned requirements are not met, a given point can be assigned to the feature consisting of less than five points, which is too small to have parameters of the model equation. In that case, the newly added point must only be closer than 1 m to any of these points. If that condition also cannot be met, a new planar feature is created, to which new points might be added during the processing of the rest of the scan. Thanks to this approach, new features are created only when a given point cannot be matched to any existing one, which results in a map with fewer features that have a larger size. It is worth noting that the described parameters were determined by conducting multiple simulation tests and choosing values that give the smallest localization error.

#### 3.2.3. Updating and Deleting Features

Since modern laser scanners provide tens of thousands of points in a single scan, it is necessary to reduce the number of created features that are stored in the map. For that purpose, co-planar and co-linear features are merged and too small ones are deleted after each scan has been processed. To further improve system performance, the number of points in each feature is decreased using a voxel grid filter. Steps performed to update all features are shown in Figure 7.

Firstly, the algorithm iterates all features to search for planes that have less than five points and lines with less than three points. These features are marked to be deleted, as they consist of a small number of points, and the feature’s parameters were not yet computed. The next step is to apply a voxel grid filter to each feature separately. Subsequently, the algorithm checks if all remaining features are valid. For this purpose it utilizes previously calculated *planarity* and *linearity* parameters according to the Equation (Equation 9). The threshold value for which feature is considered valid is set to 80% and all features that do not satisfy this requirement will be deleted. The process of deleting selected features is conducted once at the end of the update step.

Plane and line features are created and stored every time a new scan is received. It means that the number of existing features is increasing with every scan, despite the deleting and merging steps. This results in a gradually growing computation time that is necessary to assign new points to the existing features and to find correspondences between points and features during the optimization of the pose. It was therefore necessary to implement a solution that will allow PlaneLOAM to run on long sequences in real-time. This has been accomplished by selecting features that were not used as a constraint in the optimization step and marking them as *inactive*. These features are no longer taken into consideration during new points assignment and the next optimization, but they are stored as a part of the map structure. This solution is similar to the approach used in LOAM, which divides the map into cubes and discards points that are outside the range of the scanner while searching for correspondences. However, in the case of PlaneLOAM, *inactive* features are stored only for the purpose of loop closure that is based on map features (described in Section 4.3).

#### 3.2.4. Merging Features

The last element of the map processing pipeline used in PlaneLOAM is the merging of co-planar and co-linear features. In order to perform a merge of two planes or two lines, it is necessary to assure that they overlap and also to check if the merged feature would be valid. The block diagram presenting steps used to merge two features is shown in Figure 8.

Initially, in the case of planar features, the angle α between two planes is calculated according to the equation:(13)$$\alpha =\arccos\left(n_{1}\cdot n_{2}\right),$$
where $n_{i}$ is a normal vector of the *i*-th plane.

For the linear features this equation has the following form:(14)$$\alpha =\arccos\left(l_{d1}\cdot l_{d2}\right),$$
where $l_{di}$ is the normalized line direction. The maximum acceptable angle between them is set to $10^{\circ }$. If this requirement is fulfilled, we use point-to-plane (point-to-line) distances $d_{i1}$ between points from the first feature and plane (line) of the second feature to calculate the mean residual error. An analogous error is also calculated using distances $d_{i2}$ between second feature’s points and first feature’s equation. Both errors are obtained using following formulas:(15)$$\begin{array}{c} \varepsilon_{1}=\frac{1}{N_{1}}\sum d_{i1},\;i=1...N_{1} \end{array}$$(16)$$\begin{array}{c} \varepsilon_{2}=\frac{1}{N_{2}}\sum d_{i2},\;i=1...N_{2} \end{array}$$

In order to proceed with the merging procedure, the value of these errors should not exceed 0.1 m. The visualization of angle α that is considered during merging, and distance $d_{i1}$ used to calculate matching error is presented in Figure 9A.

The next step is to determine the minimum point-to-point distance $d_{2}$ between two planes (Figure 9B) and two lines (Figure 9C). It should be smaller than 1 m in order to continue merging. This condition assures that two features either overlap or are located close to each other. In the case of lines, all mentioned distances are visualized in Figure 9C.

If all the above requirements are satisfied, the last step is to verify whether a newly created feature would be valid. This is accomplished by calculating *planarity* or *linearity* of that feature. If its value is greater than 80%, it means that merging can be performed successfully. Otherwise, the merging operation is stopped, because it would create an invalid feature that would be deleted in the subsequent iteration.

#### 3.2.5. Pose Estimation

In LOAM and the proposed PlaneLOAM, the pose estimation in the mapping thread is based on the point-to-line and point-to-plane distances (Figure 10). The difference is that in PlaneLOAM, the selected linear and planar points from the most recent point cloud are matched to the globally consistent map of the high-level features. Each high-level feature has its own stored equation with information about the points belonging to this feature.

One of the challenges of using a map composed of high-level features is a fast way of matching a point to the growing number of features stored in the map. Using a linear search would not provide real-time performance for large maps. Therefore, when a point is matched to the features in the map, we firstly determine the closest point in the map using efficient kd-trees and then we retrieve the ID of the high-level feature that contains this point. Therefore, the PlaneLOAM system simultaneously manages two representations of the environment map: a point-level map like in LOAM and a map based on high-level features. This point-to-point matching to determine point-to-feature correspondence is a critical step in obtaining a system that works in real-time.

In the PlaneLOAM system, correspondence between the point and the feature in the map is verified similarly to the process of creation of features described in Section 3.2.2. The only difference is the value of the used parameters. We check that the distance to an existing plane or line ($d_{1}$) must be smaller than 0.2 m or 0.4 m accordingly, the distance to the nearest point included in the feature ($d_{2}$) must be smaller than 0.2 m, and the distance to the nearest point in the second-closest feature ($d_{3}$) must be greater than 0.7$d_{2}$.

The optimization problem in the mapping stage can be formulated as:(17)$$T^{*}=\underset{T}{\mathrm{argmin}}\left(\sum_{i}f\left(p_{i},T\right)+\sum_{j}g\left(p_{j},T\right)\right),$$
where $f\left(p_{i},T\right)$ represents the point-to-line distance while $g\left(p_{j},T\right)$ stands for the point-to-plane distance. Notice, that in contrast to the odometry stage, each point used has the same timestamp as the points are transformed to the end of the scan and thus the individual point’s acquisition time is no longer necessary to perform computation. In contrast to LOAM, in PlaneLOAM parameters of the plane and line equations used to compute point cost functions $\left(f\left(p_{i},T\right)\right.$ or $\left.g\left(p_{j},T\right)\right)$ are stored in the high-level features.

#### 3.2.6. Parameters in Mapping

Values of the parameters that are considered during adding new points to existing features, such as point-to-plane, point-to-line, and point-to-point distances, were determined experimentally, and they can be changed if the environment changes substantially with respect to the size and distribution of features (e.g., an indoor environment and a scene with a lot of vegetation in the background). The initially chosen values were updated in an iterative manner based on the localization error obtained during the evaluation of several sequences.

An analogous approach was used in the case of parameters that are used in the process of finding correspondences between the current scan points, and features stored in the map. The geometric interpretation of these parameters is the same as while creating the features, however, their initial values are more restrictive in order to ensure high certainty of finding and matching the correspondences that are used in the optimization step. Values of all the remaining parameters, such as: minimal number of points used for planar/linear features, maximal angle and minimal distance between features considered for merging, and planarity/linearity threshold, are less crucial and their values were determined by an expert only once, at the initial stage of PlaneLOAM implementation, and are constant through all datasets and experiments.

## 4. Loop Closing

The processing of the loop closing module is divided into the loop closing detection and correction of the information stored in the PlaneLOAM system, which is performed either with the pose graph representation or using optimization on high-level map features. The following sections describe our approach in more detail.

### 4.1. Loop Closing Detection

Loop closing is done using SegMap [8] system. The system is fed with laser scans with the compensated motion from PlaneLOAM to build a map of point cloud segments, where each segment has a descriptor attached. Those segments are then matched between the current local map and the global target map by comparing descriptors to recognize previously visited places. Having matched segments, SegMap computes a transformation between the sensor’s pose in the local map and the sensor’s pose at the time of the previous visit. This transformation is then used as a constraint during the optimization. An exemplary visualization of the maps build by SegMap and matches between segments is presented in Figure 11.

To produce transformations as precise as possible, we used distance threshold in the geometric consistency grouping equal to 0.5 m and a minimum cluster size of 8 (see [8] for more information about the thresholds). We further refine the transformation using a point-to-plane ICP algorithm and apply distance and cluster size threshold again to filter out inaccurate loop closures. The trajectory for global map building is computed using only ICP factors, without odometry factors. It is worth mentioning that DNN used for segments’ descriptor generation was trained on a significantly different dataset (KITTI) where not only the environment is different, but also a different type of laser scanner is used (Velodyne HDL-64E vs. Ouster OS1-64).

### 4.2. Pose Graph Optimization

The result of the odometry and mapping processing is a new pose estimate for the most recent sensor scan that increases the length of the whole trajectory. In LOAM, once the pose processing in the mapping thread is finished, the pose estimate is never updated. In PlaneLOAM, the information from the SegMap module is used to detect loop closures and update the sensor poses stored in the history. In this configuration, we adopt the typical pose graph representation (Figure 12), in which each sensor pose is represented as a node in a factor graph that gets optimized, while relative transformations between the consecutive poses are represented as edges joining these nodes. Once a new loop closure is detected by SegMap, a new relative transformation (an edge) between the corresponding nodes in the graph is added. During the optimization, new pose values are sought that minimize the following error:(18)$$\underset{T_{1},...,T_{n}}{\mathrm{argmin}}\left(\sum_{i}h\left(T_{i},T_{i+1}\right)\Omega_{i,i+1}h\left(T_{i},T_{i+1}\right)+\sum_{j}h\left(T_{l(j,1)},T_{l(j,2)}\right)\Omega_{l(j)}h\left(T_{l(j,1)},T_{l(j,2)}\right)\right),$$
where $T_{1},...,T_{n}$ are the optimized sensor poses, $h(T_{i},T_{i+1})$ is the pose error between the *i*-th and (i+1)-th poses from the measurement of the consecutive poses (obtained from the mapping step), and $h(T_{l(j,1)},T_{l(j,2)})$ is the pose error of the *j*-th loop closure measurement defined as the error between l(j,1)-th and l(j,2)-th poses related by that loop closure observation.

In the case of pose graph optimization, direct observations of the features are not utilized, as these measurements are used only to compute the pose-to-pose constraints in the graph.

### 4.3. Loop Closure Based on Map Features

The experience from the visual SLAM shows us that the best accuracy should be expected when the sensor’s poses are optimized jointly with features using bundle adjustment [7]. In PlaneLOAM, we prepared a system to perform a so-called global Bundle adjustment (GBA) that optimizes all of the available poses along with all of the available parameters of the features to determine the most accurate trajectory estimate (Figure 13). In the presented system, we perform GBA only after a loop closure was detected.

The SegMap module provides our system with a relative transformation between two sensor poses that are treated as initial information for loop closure considerations. In PlaneLOAM, we would prefer to utilize the feature-based structure of the map to further improve the accuracy of the initial relative transformation. Therefore, let us consider two poses *A* and *B* that were matched according to SegMap. First of all, we determine the set of plane and line features visible from poses *A* and *B*, denoting them by $F_{A}$ and $F_{B}$. Then, based on the SegMap relative transformation, we perform matching of the points in features $F_{A}$, associating them to the features $F_{B}$. The matching process follows the same set of rules as in the mapping module and is followed by the optimization typically used at that stage. As a result, we obtain an optimized, corrected transformation between the poses *A* and *B* that is used for further processing in the loop closure pipeline.

This, already accurate, transformation is used to merge high-level map features that represent the same physical plane/line but were assigned different IDs due to the drift of the system. Once features are merged according to the already described rules of feature merging, the GBA optimization is performed:(19)$$\underset{T,P,L}{\mathrm{argmin}}\left(\sum_{i=1}^{n}\sum_{j\in P_{i}}f\left(p_{i},T_{j}\right)\Omega_{i,j}f\left(p_{i},T_{j}\right)+\sum_{i=1}^{n}\sum_{j\in L_{i}}g\left(p_{i},T_{j}\right)\Omega_{i,j}g\left(p_{i},T_{j}\right)\right),$$
where T is a set of all optimized sensor poses, P is the set of all optimized planes, L is the set of all optimized lines, $P_{i}$ and $L_{i}$ define the sets of indices of all planes and lines respectively visible from *i*-th sensor pose, $f\left(p_{i},T_{j}\right)$ is the error function of the measurement between the *i*-th plane and *j*-th pose, and $g\left(p_{i},T_{j}\right)$ is the error function of the measurement between the *i*-th line and *j*-th pose.

During the optimization, the sets of parameters for the sensor poses, plane equations, and line parameters are simultaneously changing their values to reduce the total error of the constraints in the system. This stands in contrary to LOAM that computes plane and line equations when needed and to PlaneLOAM operating without loop closures, which alternates the pose estimation and line/plane equation estimation steps.

Including the parameters of the plane and line parameters directly into the optimization is feasible under two assumptions: the minimal representation of features is used and the Jacobians with respect to the plane/line parameters are well defined. Therefore, for the time of the optimization, the parameters of planes and lines are converted into the proposed minimal representations, then the optimization is performed, and the obtained results are transformed back to the original representations to be used to update the poses and the map of the system. The following paragraphs introduce in more detail the used minimal representation for both planes and lines.

#### 4.3.1. Plane Representation for Optimization

The parameters of the planes in PlaneLOAM are represented by the four-dimensional vector $\left[\begin{array}{cc} n & n_{d} \end{array}\right]$, where n is the three-dimensional plane normal, and $n_{d}$ is the distance to the origin of the coordinate system. The minimal representation for planes is 3-dimensional as the plane normal vector is normalized. In order to devise a minimal representation, we opted to use the ideas presented in [44], where the exponential and logarithmic functions map elements from the (n + 1)-dimensional spheres to n-dimensional tangent hyperplanes. Using this concept, we can assume our plane normal $\left(n_{3\times 1}={\left[\begin{array}{ccc} n_{1} & n_{2} & n_{3} \end{array}\right]}^{T}\right)$ to be an element of $S^{2}$ group sphere and we can map it to the point on the tangent hyperplane $\left({\left[\begin{array}{cc} \omega_{x} & \omega_{y} \end{array}\right]}^{T}\right)$:(20)$$\begin{array}{cc} \theta & =\mathrm{acos}(n_{3}), \end{array}$$(21)$$\begin{array}{cc} \omega_{x} & =-n_{2}\frac{\theta }{\sin(\theta )},\;\;\;\omega_{y}=n_{1}\frac{\theta }{\sin(\theta )}, \end{array}$$
where ${\left[\begin{array}{cc} \omega_{x} & \omega_{y} \end{array}\right]}^{T}$ fully encode our information about the plane normal n. As a result, the minimal, three-dimensional representation of the plane equation ($\omega_{x},\omega_{y},n_{d}$) is computed from the original, four-dimensional representation ($n,n_{d}$). Based on these equations, it is possible to compute the corresponding analytical Jacobian with respect to each component of the proposed representation.

This spherical conversion has two special cases that need to be taken care of. The first case occurs when θ→0 that requires a series expansion of $\frac{\theta }{\sin(\theta )}$. The second case is when θ→π that we completely avoid by computing the results for an equivalent plane representation of $n^{\prime }=-n$ and ${d_{n}}^{\prime }=-d_{n}$.

Once optimized, the native, four-dimensional representation $\left(n={\left[\begin{array}{ccc} n_{1} & n_{2} & n_{3} \end{array}\right]}^{T},n_{d}\right)$ can be retrieved with:(22)$$\begin{array}{cc} \theta & =\sqrt{\omega_{x}^{2}+\omega_{y}^{2}}, \end{array}$$(23)$$\begin{array}{cc} n_{1} & =\omega_{x}\frac{\sin(\theta )}{\theta },\;\;\;n_{2}=\omega_{y}\frac{\sin(\theta )}{\theta },\;\;\;n_{3}=\cos\left(\theta \right), \end{array}$$
based on the logarithmic map from the tangent hyperplane to the point belonging to the $S^{2}$ group.

#### 4.3.2. Minimal Line Representation

The lines in PlaneLOAM are represented by the six-dimensional Plücker coordinates $\left[\begin{array}{cc} l_{d} & l_{m} \end{array}\right]$, where $l_{d}$ is the direction of the line, and $l_{m}$ is the moment of the line (both three-dimensional) that are perpendicular to each other. In the proposed implementation, we keep the direction of the line $l_{d}$ normalized with its first component positive $l_{d}\left(0\right)\geq 0$ to be consistent. In order to devise a minimal representation, we opted to use the ideas presented in [45], where the properties of the SO(3) group are utilized. In this approach, the information to encode is presented in the form of a SO(3) group element (rotation matrix):(24)$$R=\left[\begin{array}{c} l_{d},\frac{l_{m}}{∥l_{m}∥},\frac{l_{d}\times l_{m}}{∥l_{m}∥} \end{array}\right],$$
and then used to compute the representation in the Lie algebra:(25)$$\omega =\log\left(R\right),$$
where log(·) computes the Lie algebra representation for the Lie group element R. The ω encodes the information about the normalized direction of the line and the normalized moment of the line. To be able to fully reconstruct the original line parametrization we store the length of the line moment ($l_{m}$) as the fourth parameter ($m=∥l_{m}∥$). Summarizing, the original representation (${\left(l_{d},l_{m}\right)}_{6\times 1})$ is converted into its minimal form ${\left(\omega ,m\right)}_{4\times 1}$. Based on the presented equations, it is also possible to compute an analytical Jacobian with respect to each component of the proposed representation.

For the minimal line parametrization we have three special cases:$∥\omega ∥\to 0$—we use a series expansion of $\frac{\theta }{\sin(\theta )}$;$∥\omega ∥\to \pi$—we avoid the issue by performing computation with equivalent Plücker representation (${l_{d}}^{\prime }=-l_{d}$ and ${l_{m}}^{\prime }=-l_{m}$);$∥l_{m}∥=0$—we set m=0 and follow the conversion with a new random, unit-length vector of $l_{m}$ that is perpendicular to $l_{d}$. As m=0, the random choice of $l_{m}$ has no impact on Plücker coordinates obtained from the minimal representation.

Once optimized, the native, six-dimensional representation can be retrieved with:(26)$$l_{d}=\exp\left(\omega \right)\left[\begin{array}{c} 1\\ 0\\ 0 \end{array}\right],\;l_{m}=m\exp\left(\omega \right)\left[\begin{array}{c} 0\\ 1\\ 0 \end{array}\right],$$
where exp(·) computes the Lie group representation based on the Lie algebra element.

## 5. Experimental Evaluation

### 5.1. Accuracy of Trajectory Estimation with High-Level Features

We performed a series of experiments in order to quantitatively evaluate the accuracy of trajectory estimation with PlaneLOAM. As the aim of these experiments was mainly to determine if the more elaborated structure of the global map contributes to the accuracy of trajectory estimation, we compared PlaneLOAM results to the results we got with the same sequences using the publicly available open-source version of the LOAM system (This version was obtained from https://github.com/laboshinl/loam_velodyne accessed on 25 January 2019). The evaluation and comparison methodology is based on the absolute trajectory error (ATE) metric introduced in [46] and widely used to evaluate SLAM systems. The ATE metric determines how far from the given pose of the sensor its counterpart is located on the estimated trajectory by computing the Euclidean distance between the corresponding points of the ground truth and the estimated trajectory. These correspondences are determined with the time stamps established while acquiring the scans. Before ATE is computed, it is necessary to transform both trajectories to the common reference frame by aligning them together. We applied the Umeyama algorithm [47] to compute the rigid body transformation between the two sets of points (i.e., sensor poses) representing the trajectories. All estimated poses are used to compute the alignment. Although it is possible to use only a few initial poses to compute the transformation, it results in an increase of the ATE errors over time, therefore we prefer to compute a rigid transformation of full trajectories. This alignment yields a less intuitive distribution of errors over time but also makes the error metric for the whole trajectory less dependent on the moments when larger estimation errors have occurred. This is particularly important in the context of orientation errors that tend to propagate to large pose differences if only the initial states are used for alignment. To obtain quantitative results we compute the root mean squared error (RMSE) of the ATE metric for the whole trajectory $\left({\mathrm{ATE}}_{\mathrm{RMS}}\right)$, showing also the maximum local ATE values for particular estimated trajectories $\left({\mathrm{ATE}}_{\max}\right)$ and the standard deviation over trajectory $\left(\sigma_{\mathrm{ATE}}\right)$. Larger values of $\sigma_{\mathrm{ATE}}$ suggest that the investigated method has problems with particular parts of the trajectory, while performing better elsewhere.

The experiments involved two publicly available data sets: the KITTI odometry benchmark [48] and the MulRan multimodal range dataset [49]. The KITTI benchmark, widely used to test SLAM algorithms [21,32,39,41], contains 11 sequences with LiDAR data and ground truth trajectories. A Velodyne HDL-64E LiDAR (64 channels) mounted on a car was used to collect the scans, while the ground truth poses were obtained using GPS and inertial navigation. Although by default the KITTI benchmark has the motion distortions in LiDAR scans corrected using an undisclosed algorithm, we used the raw scans in our experiments. Considering that PlaneLOAM is intended for environments with geometric structures we selected for evaluation three KITTI sequences: 00, 05, and 07, which were acquired in residential areas, thus they contain many planar and linear structures in the environment. These sequences contain also loops, making it possible to evaluate the benefits of the feature-based approach to loop closing. The recently published MulRan dataset is mainly intended for testing place recognition, but it contains accurate ground truth trajectories, which makes it possible to use it also for SLAM evaluation. MulRan contains sequences collected in four environments, among which we selected the Dajeon Convention Center (DCC), which is structurally diverse, with narrow roads between high-rise buildings and with several loops performed by the vehicle to allow testing our loop closing methods. The LiDAR used in MulRan is a recent Ouster OS1-64 (64 channels). The ground truth trajectories were made via multi-sensor SLAM using inertial sensors, GPS, and ICP scan matching. Although MulRan contains multi-sensory data we explored only the raw laser scans and ground truth trajectories.

The accuracy of the KITTI 00 trajectory estimation for PlaneLOAM and the open-source LOAM version is visualized in Figure 14A,B, respectively, using the graphical convention of ATE plots introduced by Sturm et al. [46]. As LOAM does not have loop detection and closing capabilities, for this test we did not use the loop closing procedure in PlaneLOAM, which allows us to compare the accuracy for two systems using similar scan-to-scan plus scan-to-map registration architectures. The main difference is the use of high-level features in PlaneLOAM, which resulted in a more accurate trajectory and smaller ATE residuals (Figure 14C). In our tests, LOAM achieves similar LiDAR odometry performance on the KITTI sequences as the results reported in other works [8,25] using the open-source version.

It is worth noting that a better representation of the ground plane in PlaneLOAM allowed our system to keep more accurate elevation estimates for the whole trajectory (Figure 14D). Quantitative results for the KITTI 00, 05, and 07 sequences are provided in Table 1. In all these cases of well-structured urban environments, our approach outperforms the open-source LOAM with respect to the ATE metric, considering the RMS and maximum error values, achieving also a smaller standard deviation of the estimates.

Visualizations of selected fragments of the global maps of high-level geometric features produced by PlaneLOAM are shown in Figure 15. Maps for the KITTI 00 sequence are shown in Figure 15A,B, while Figure 15C depicts a fragment of the map obtained for the sequence 07. For the sake of clarity, only planar features are visualized. Horizontal planar features (the ground plane) are orange, larger approximately vertical features (facades, walls, fencing, etc.) are yellow, while smaller vertical features (those containing less than 50 scanned points) are shown in cyan.

A similar experiment was performed for the Mulran DCC sequence. The trajectories with ATE values estimated by LOAM and PlaneLOAM are shown in Figure 16A,B, respectively. For this long sequence acquired in a diversified environment, the errors were bigger than in the KITTI tests for both investigated SLAM systems, however, PlaneLOAM again estimated a more accurate trajectory, achieving also smaller ATE values (Figure 16C) and better elevation estimation (Figure 16D). Quantitative results of this experiment are summarized in Table 1.

### 5.2. SLAM with High-Level Features in Different Environments

As an important aspect of our research is to demonstrate the practical applicability of our approach to LiDAR SLAM with different 3-D laser scanners and in scenarios typical to the localization of vehicles in urban traffic, e.g., for city buses, we tested PlaneLOAM also on sequences from our own experiments. One of these sequences was acquired using a car with a roof-mounted Sick MRS-6124 LiDAR (24 channels, 120${}^{\circ }$ forward-looking field of view) and a differential GPS (DGPS) Ublox C099-F9P/ZED-F9P module. This RTK-GPS achieves centimeter-level accuracy of positioning at 10 Hz, whenever it observes enough satellites. This setup was used in a mixed outdoor/indoor experiment scenario implemented in the large shopping mall Posnania in the city of Poznan. The car started from a public road nearby Posnania, then approached the underground parking lot entrance, traversed the underground area, and came back to a different public road on the other side of the large building. Notice that in this experiment the ground truth from GPS was recorded only for the outdoor part, however, we intended to demonstrate that PlaneLOAM can handle such mixed scenarios and resumes tracking of the ground truth trajectory with reasonable errors after traveling through the undergrounds.

The positional errors with respect to the ground truth trajectory for PlaneLOAM and LOAM estimates are depicted in Figure 17A,B, respectively. From the plot of ATE values vs. time (Figure 17C) it can be observed that LOAM has a significantly larger ATE for most of the trajectory. Although the largest error at the start of the sequence is a result of trajectory alignment performed by the script from [46] that computes the ATE metric and generates the plot, this indicates that LOAM had a large orientation error in the first part of the trajectory and aligning the trajectories using only the initial part would result in an even bigger discrepancy in the remaining part of the plot. However, we are interested in absolute pose errors with respect to an external reference system, as a trajectory that has small ATE for consecutive poses of the sensor implicitly shows that the global map estimated jointly with this trajectory is also correct. This consistency is not caught by relative error metrics, as the one used by Geiger et al. [48] or the RPE (relative pose error) defined together with ATE in [46]. In order to demonstrate that the global maps produced by PlaneLOAM are topologically consistent and may be used for such tasks as motion planning, we show in Figure 18A the entire map estimated by PlaneLOAM in the Posnania experiment. As it is visible in Figure 18B the map is rich in geometric details, correctly localizing even small architectural elements, such as pillars, despite the limited scanning density and horizontal field of view of the Sick LiDAR.

The second experiment in this series was implemented using a city bus equipped with the same Sick MRS-6124 LiDAR and a GPS receiver (Figure 19A). Several sequences were collected in a small town nearby Poznan, characterized by a moderately structured environment with quite narrow roads surrounded by residential buildings, but also many trees and some bushes along the roads (Figure 19B). The rather low-rise buildings located around made it possible to have a good GPS signal all the time. The small town experiments have been conducted in February during cloudy weather with occasional rain, however, these conditions apparently did not affect the results of the Sick LiDAR measurements.

We show the estimated trajectories from both PlaneLOAM and LOAM with respect to the ground truth on the same plots for the TownCentre (Figure 19C) and TownSuburbs (Figure 19E) experiments, without visualization of the ATE values, in order to keep the figure compact. The ATE results are plotted in Figure 19D,F for TownCentre and TownSuburbs, respectively. As it can be seen, PlaneLOAM outperforms LOAM in both sequences, although the environment was less structured than in the Posnania case and mounting the LiDAR high on the bus roof made the sensor motion less smooth due to vibrations. Statistics of the ATE metric are provided in Table 2 for all three scenarios.

### 5.3. Analysis of the Computation Time

The new map structure proposed in PlaneLOAM implies changes in the computational load of the operations that are necessary to process a single scan. These changes are insignificant in the laser odometry thread, which is largely inherited from the open-source LOAM, but are important in the mapping thread, which is responsible for the final, more accurate pose estimates. Therefore, we provide a detailed analysis of the computation time spent by particular processing steps of the mapping thread in PlaneLOAM and the open-source LOAM, which is considered a baseline. This side-by-side analysis and comparison, presented in Table 3, demonstrates that in spite of the more complicated map structure, the high-level geometric features make it possible to decrease the processing time, mostly because the number of individual elements that have to be searched and compared is considerably smaller.

The computation time analysis was performed on an Intel Core i7-9700TE PC, using the Mulran DCC sequence, which we consider the most demanding one among those used in the paper, because of large point clouds yielded by the Ouster OS1-64 LiDAR, and the diversified environment that generated both planar and line features. The time values in Table 3 are averaged over the 5542 scans in the Mulran DCC sequence. The observed standard deviation of the computation time was insignificant, and in PlaneLOAM depended mostly on the number of features generated per scan. The PlaneLOAM processing steps listed in Table 3 are described in Section 3.2, while a reader interested in their counterparts in LOAM should consult [9].

The total processing time for PlaneLOAM is slightly smaller than for the open-source LOAM, mostly due to the much shorter time PlaneLOAM spends on creating the kd-tree. PlaneLOAM employs local kd-trees to find points used to compute the point-to-feature constraints in optimization but does not use this method while creating and updating the features. In the case of PlaneLOAM, the kd-tree contains only the points that constitute the high-level features, thus the number of points being considered is much smaller than in LOAM. Moreover, in PlaneLOAM the features that are inactive at the given time instance are not considered for any matching operations, which conserves time when computing the point-to-feature constraints.

The comparison of computing time does not include the loop closing module, which is not present in the open-source LOAM. The inference time for the SegMap method that finds the loop closures is 155 ms on average for the same Mulran DCC sequence, while optimization of the pose graph using g${}^{2}$o takes from 60 to about 240 s, depending on the number of poses included in the graph. These values suggest that although loop closures based on the pose graph cannot be computed in real-time, the time delays of few minutes are still acceptable for on-line operation, providing that these loop closures are computed in a separate thread running in background. The full factor graph optimization is even slower and can take from 280 to 500 s for the sequences considered in the paper. However, full optimization takes into account multiple loop closures and is computed off-line, as this PlaneLOAM module is aimed at producing a very accurate metric map of the environment that can be then used for motion planning.

### 5.4. Different Approaches to Loop Closing in Feature-Based LiDAR SLAM

The third series of experiments concerns the integration of the SegMatch/SegMap loop detection method with PlaneLOAM. Most LiDAR SLAM systems with loop detection close the loops using a pose graph approach, which creates edges between pose nodes using some form of motion constraints (relative transformations) between the associated locations, e.g., applying ICP [14] or directly using the SegMatch method [30]. In contrast, we aim to demonstrate that the map representation using high-level features opens a possibility to treat the detected loop candidates in a way similar to the loop closure approach typical to visual SLAM, i.e., by associating the individual features and then optimizing the factor graph in a BA-like manner.

Although there is a number of techniques that can be used to detect the loop closing event using LiDAR range data (see [15] for a brief experimental comparison), we believe that they are either inferior to the applied SegMap method, or require a particular map representation [20,21], that cannot be implemented in PlaneLOAM. Hence, in this paper we do not compare different techniques that detect the re-visited areas but focus on the comparison of the approaches to exploit the detected loop closure in order to improve the accuracy of the trajectory and consistency of the map.

The sequences chosen to evaluate the approaches to loop closing are KITTI 00 and MulRan DCC as both contain multiple loops and represent different environments and LiDAR sensors (Velodyne and Ouster, respectively). The resulting trajectories are plotted in Figure 20. Two approaches to loop closing are compared: a simple method based on the pose graph produced by SegMap (Figure 20B,E) and our approach based on matching the high-level features and optimization of the resulting factor graph using the g${}^{2}$o library (Figure 20C,F). To make the gains due to loop closures more evident we show in the same figure also the trajectories estimated without loop detection and closing (Figure 20A,D).

The pose-based approach is similar to the one presented in [8], as it uses either PlaneLOAM or LOAM for estimating the LiDAR odometry between the consecutive sensor poses (graph nodes) and for motion compensation in the 3-D scans, while loop detections are found using SegMap algorithm. Loop detections are added as constraints to the pose graph to be optimized. The BA-like approach was applied only in PlaneLOAM, as the original LOAM does not have a map structure that supports feature-level matching and optimization. The trajectories estimated by LOAM and PlaneLOAM without loop closing are presented in Figure 20A,D for KITTI 00 and MulRan DCC, respectively. Figure 20B,E show the trajectories estimated using the pose-based approach to loop closing, also for KITTI 00 and MulRan DCC, respectively. The BA-like approach is presented only for PlaneLOAM, for KITTI 00 in Figure 20C, and for MulRan DCC in Figure 20F. It is apparent, that closing the loops with the pose-based approach brings gains with respect to the open-loop version for both LOAM and PlaneLOAM. The improvement in terms of ATE RMS in the case of KITTI 00 is about 13% for PlaneLOAM and 28% for LOAM, as shown by the quantitative results in Table 4. Closing the loops brings more gain to the LOAM accuracy, as the baseline results without loop closing were worse. However, PlaneLOAM achieves better absolute results in all the considered variants. In the MulRan DCC case (Table 5) the gain due to loop closures with the pose-based variants is bigger. It is about 32% for PlaneLOAM and 34% for LOAM. The DCC scenario is more diversified than the KITTI 00 sequence with respect to the environment features, thus being less favorable to mapping with high-level features, which explains a smaller difference between PlaneLOAM and LOAM. However, PlaneLOAM still has the smallest absolute error values in all cases.

The gain in trajectory estimation accuracy is considerably bigger if loop closing is implemented using the features matching capability of PlaneLOAM and BA-like optimization, which corrects not only the sensor poses, but also the locations of features. The gain with respect to the baseline approach for KITTI 00 is about 45%, while for the MulRan DCC the improvement reaches 51% in terms of ATE RMSE. This is consistent with previous results, as PlaneLOAM performed better on KITTI than on MulRan, thus the improvement due to loop closures may be bigger. However, it should be noted that the environment representation based on high-level features turned out to be appropriate for computing loop closures even in the MulRan DCC case, where the detection of loops as unique locations re-visited by the vehicle was to a large extent dependent on such objects as trees and other vegetation. Whereas these elements of the scene create a context that is useful for place recognition, observations of the remaining geometric structures provide enough constraints to improve the estimated trajectory.

The loop closure method exploiting BA-like optimization manages to close many loops of different sizes in a single optimization session. This is beneficial whenever the test vehicle traverses a closed path and re-visits a given location several times. Such a situation occurred in the Mulran DCC sequence. The vehicle visited the same location (pointed by the arrow ➀ in Figure 20F) four times, closing loops of different sizes: a smaller loop, when traversing the inner, nearly circular path, and a much bigger one, when traversing the outer path encircling the entire DCC area. The loop closures were successful in spite of the fact that the vehicle traversed the loop closure venue moving in different directions, as depicted by the example Ouster OS1-64 point clouds in Figure 21A,B. The loop closure module tolerates also moderate changes in the environment appearance that occur in-between the re-observations of loop closure venues, as demonstrated by another example taken from the Mulran DCC experiment and shown in Figure 21C,D. The location where this situation has occurred is pointed by the arrow ➁ in Figure 20F. Both results corroborate the claim that the combination of location matching using learned SegMap descriptors, and BA-like optimization using features create a robust and versatile loop closure mechanism that is applicable to practical scenarios in autonomous driving and large-scale environment mapping.

Finally, we compared the trajectory estimation results for the KITTI 00 sequence under the different approaches to loop closures using the evaluation method originally proposed with the KITTI dataset [48]. Unlike the ATE metrics, this method computes separate translation and rotation errors relative to the covered distance and presented as a function of the trajectory’s segment length. Whereas these relative accuracy metrics do not translate directly to inaccuracies in the estimated trajectory shape, they are consistent with the results presented in [8], where both translation (Figure 22A) and rotation (Figure 22B) errors reduce over longer paths, as for short paths the results depend on the open-loop LiDAR odometry. The translation error plots have a similar decreasing tendency for longer paths in the case of both LOAM and PlaneLOAM using the pose-graph approach, but PlaneLOAM achieves smaller error values due to better odometry estimates. The approach to loop closing fully exploiting the PlaneLOAM high-level features makes it possible to achieve even better results, but only for longer paths, as for the short ones it uses the same LiDAR odometry as the baseline variant. In this context, it should be noticed that the detected loop closures are not evenly distributed along the path. As the vehicle covers a longer distance, the number of detected loop closure candidates increases considerably (Figure 22C), because the vehicle returns to the already mapped areas.

## 6. Conclusions

The feature-based system has the potential of improving accuracy as it reduces error during data association. Features are created using a greater number of points, which translates into better precision of calculated plane equations.

Conducted experiments showed that as far as accuracy is considered, the modified system achieved better results in all recorded sequences. The proposed system is also characterized by a lesser number of points that are used for pose optimization. This is due to the fact that constraints are calculated in a more restrictive manner. Thanks to that, correspondences are computed more precisely, providing better accuracy and results. Besides evaluation on the publicly available datasets, experimental tests were performed at the PUT campus and on public roads, in order to verify the PlaneLOAM performance in real-world conditions. These tests confirmed also that it is possible to use our feature-based approach with LiDAR for localization and structure-rich mapping in large indoor environments.

It is worth mentioning that the developed system performance substantially depends on selected parameters’ values, as they have a significant influence on the size and number of created features. Therefore, inappropriately selected ones might cause the system to work unreliably. For this reason, a significant amount of time was spent to select universal parameters’ values, that were finally utilized and are suitable for different types of environments.

Although the PlaneLOAM mapping thread is even slightly faster than its counterpart in the open-source LOAM, the processing time can be further decreased by code optimization. For instance, operations on LiDAR points can be shortened using the kd-tree algorithm while creating and updating the high-level features. Code optimization is not the only direction of PlaneLOAM future development. We are concerned with two aspects of its architecture: more efficient map management for optimization and robustness to more cluttered and dynamic environments. One of the considered solutions is to introduce a keyframe idea, already used in visual SLAM [5] that would reduce the number of optimized poses, and thus increase the processing speed of pose graph optimization. Similarly, in the case of global BA, it is possible to achieve significant time processing gains by applying a marginalization technique when solving the bundle adjustment problem. Explicit elimination of non-stationary objects (e.g., cars, pedestrians) from the processed data would be possible using our recently published deep learning-based method [50], which was already tested with PlaneLOAM.

## Figures and Tables

**Figure 1 sensors-21-03445-f001:**
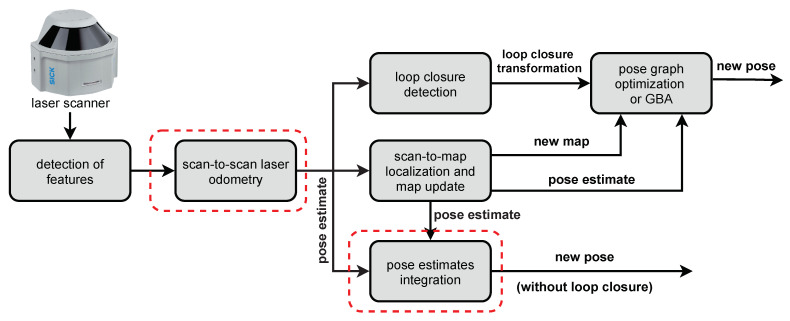
Block scheme of the general PlaneLOAM architecture. Blocks in dotted-line borders do not contain any significant changes in relation to LOAM.

**Figure 2 sensors-21-03445-f002:**
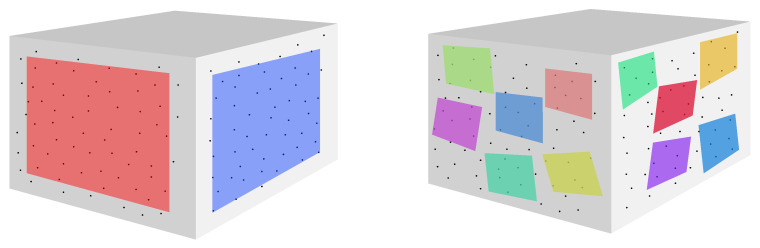
A conceptual comparison between the high-level planar feature in the PlaneLOAM system (**left**) and the cloud of planar points in the LOAM system (**right**).

**Figure 3 sensors-21-03445-f003:**
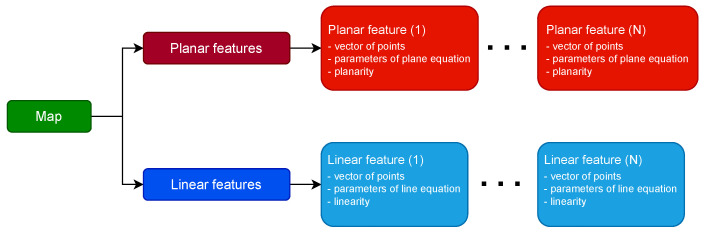
Structure of stored environment map consists of lines and planes forming high-level features.

**Figure 4 sensors-21-03445-f004:**

Processing pipeline implemented in the system, consisting of creating, updating, deleting and merging features.

**Figure 5 sensors-21-03445-f005:**
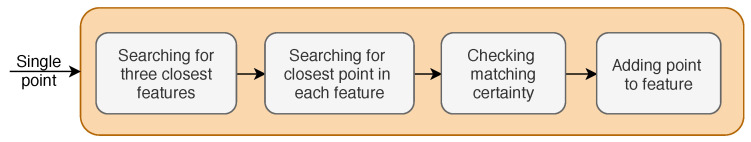
Steps performed to match a single point to an existing feature.

**Figure 6 sensors-21-03445-f006:**
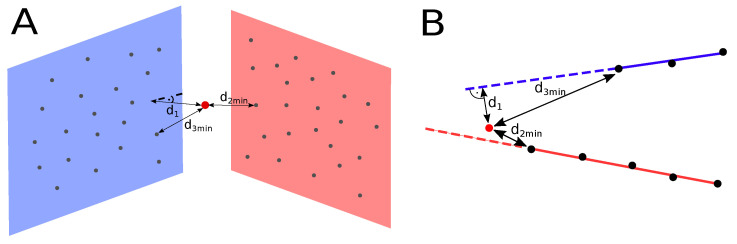
Distance to the closest plane ($d_{1}$), distance to the closest point ($d_{2}$), and the distance to the second closest feature ($d_{3}$), which are considered during the process of addition of a point to the existing planar (**A**) and linear (**B**) features.

**Figure 7 sensors-21-03445-f007:**
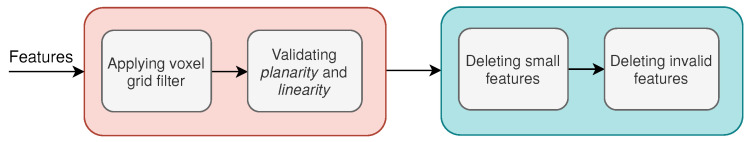
Steps performed to update existing features and delete too small or invalid ones.

**Figure 8 sensors-21-03445-f008:**
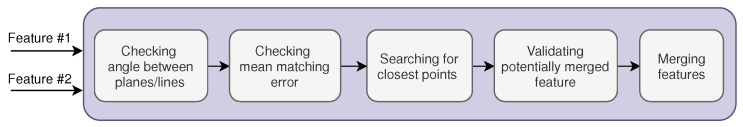
Steps performed to merge two features, based on angle, matching error, and distance between them.

**Figure 9 sensors-21-03445-f009:**
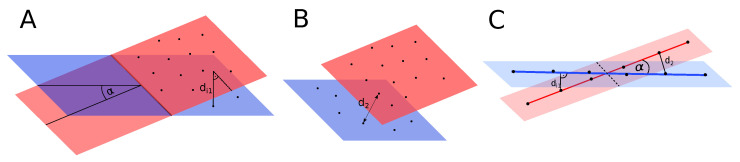
Angle between two planes (α) and exemplary point-to-plane distance ($d_{i1}$) (**A**), distance $d_{2}$ between two points that belong to planes considered for merge (**B**), angle between two lines α, an exemplary point-to-line distance $d_{i1}$ and distance between two points that belong to lines considered for matching $d_{2}$ (**C**).

**Figure 10 sensors-21-03445-f010:**
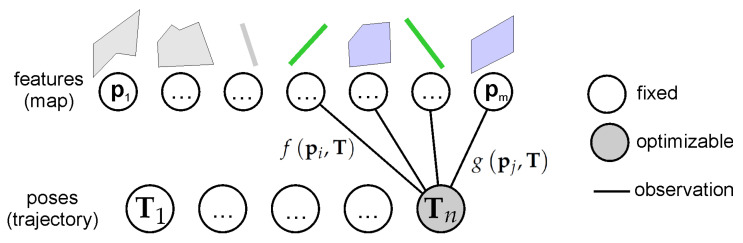
The optimization problem in PlaneLOAM mapping is solved by computing the current pose T using the point-to-line ($f(p_{i},T)$) and point-to-plane ($g(p_{j},T)$) constraints. Parameters of the observed features remain fixed during pose optimization. Compared to the LOAM approach, the parameters of each feature are stored explicitly in the map.

**Figure 11 sensors-21-03445-f011:**
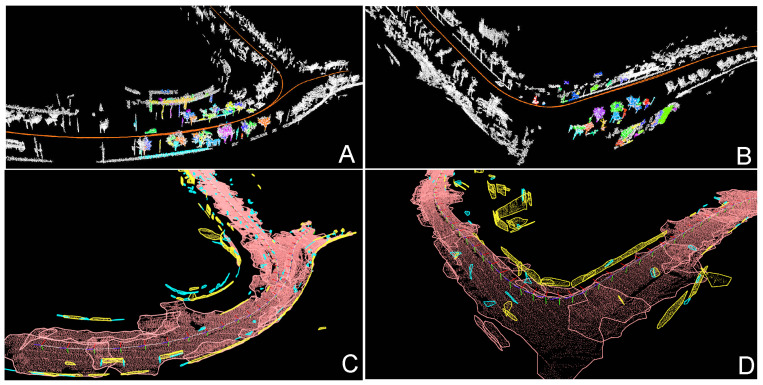
Detecting loop closures with the SegMap approach in the MulRan dataset sequence (**A**,**B**). Local maps of semantically segmented point clouds visualized in different colors are matched to a global map of point cloud segments shown in white. The same areas are visualized as maps of high-level features (**C**,**D**), with orange planar patches for the ground plane and the approximately vertical features shown in yellow (larger walls) and cyan (small planar elements).

**Figure 12 sensors-21-03445-f012:**
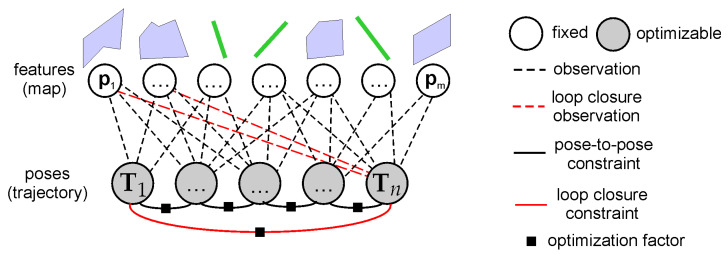
Pose graph optimization in PlaneLOAM is performed using the pose-to-pose constraints obtained from the mapping thread. These constraints join consecutive poses in the trajectory. Loop closure constraints join two distant poses, and are established upon segment-based loop closure detection. Features stored in the map are not included in optimization, as in the pose graph the observations are aggregated into pose-to-pose constraints.

**Figure 13 sensors-21-03445-f013:**
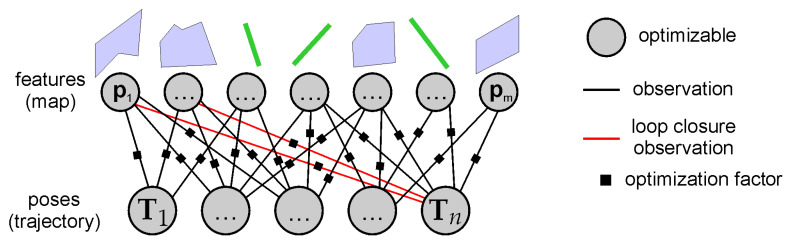
Global optimization based on map features involves optimization of the pose estimates (T) and features planes (p) using the pose-to-feature constraints. There are no pose-to-pose constraints and loop closure information is included directly by merging corresponding features from the re-visited locations.

**Figure 14 sensors-21-03445-f014:**
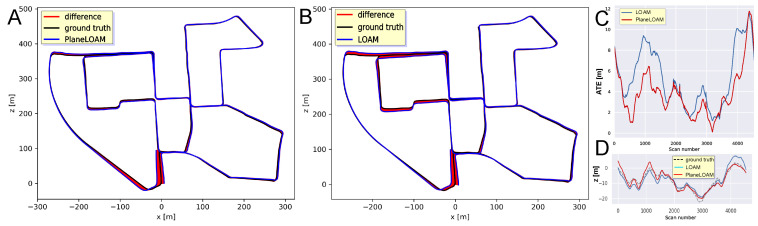
Absolute trajectory errors for the KITTI 00 sequence acquired with a Velodyne HDL-64E LiDAR. The PlaneLOAM trajectory (**A**) has smaller deviations from the ground truth than its counterpart estimated with the open-source LOAM (**B**). This is visible in the plot showing ATE errors vs. consecutive scan numbers (**C**). Smaller errors in the elevation (*z* axis) contribute to the overall better performance of PlaneLOAM (**D**).

**Figure 15 sensors-21-03445-f015:**
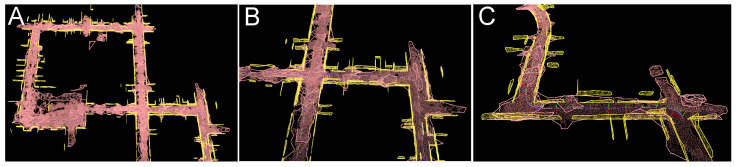
Fragments of the KITTI 00 (**A**,**B**) and KITTI 07 (**C**) global maps built by PlaneLOAM and consisting of high-level geometric features. Approximately vertical planar features are shown in yellow, while orange patches denote the ground plane.

**Figure 16 sensors-21-03445-f016:**
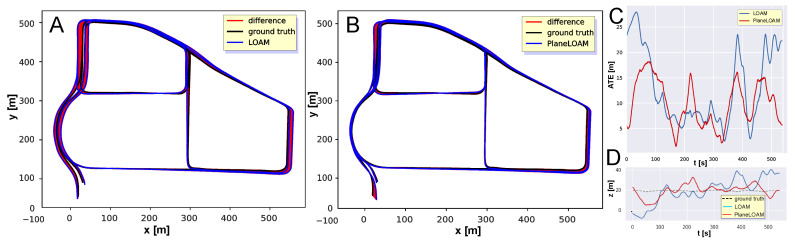
Absolute trajectory errors for the MulRan DCC sequence acquired with an Ouster OS1-64 LiDAR. This long sequence contains multiple loops, but the results of PlaneLOAM (**A**) and LOAM (**B**) estimation were obtained without any loop detection and closing. The plot of the ATE values vs. time (**C**) shows that PlaneLOAM keeps the overall pose error smaller than LOAM for most of the trajectory, and in particular it yields better estimates of the elevation values (**D**).

**Figure 17 sensors-21-03445-f017:**
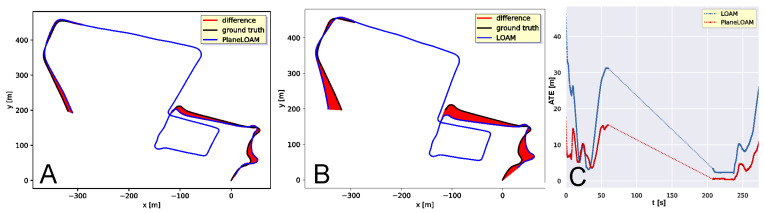
Results of an experiment in mixed outdoor/indoor environment of the Posnania shopping mall. The trajectories estimated by PlaneLOAM (**A**) and LOAM (**B**) were obtained from scans yielded by a Sick MRS-6124 LiDAR. Although in this experiment the ground truth DGPS trajectory was available only for the outdoor part, the plot of ATE (**C**) shows that PlaneLOAM was able to keep a much smaller drift while moving through the underground parking lot, which is an example of a highly structured environment.

**Figure 18 sensors-21-03445-f018:**
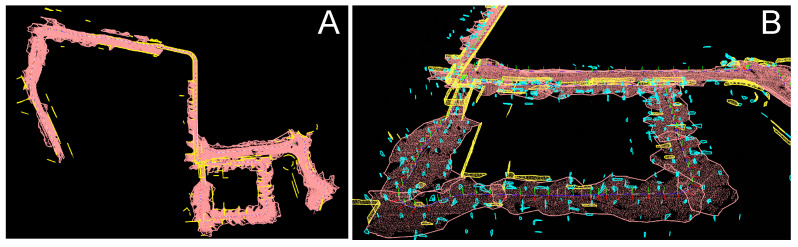
Visualization of the entire global map built by PlaneLOAM during the Posnania experiment (**A**), and a close-up view of its central fragment (**B**) showing a detailed structure of the underground parking lot with walls (yellow), smaller vertical elements, such as pillars (cyan) and the ground plane (orange).

**Figure 19 sensors-21-03445-f019:**
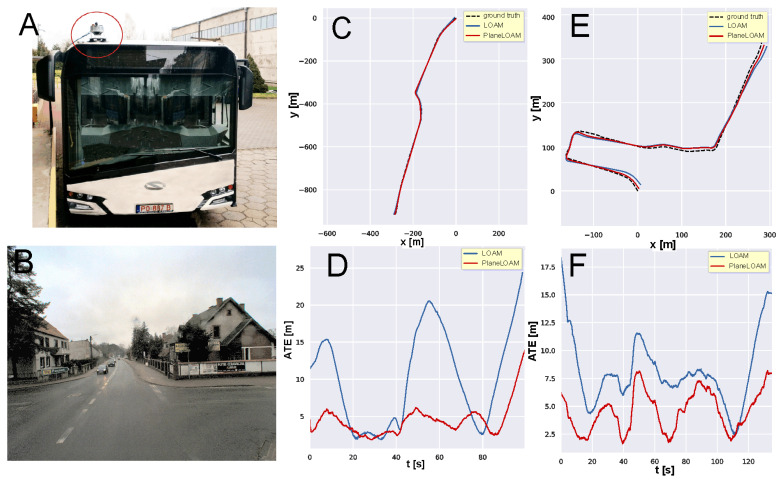
Results of experiments in localization of a city bus equipped with a single Sick MRS-6124 LiDAR (**A**). The two sequences, denoted TownCentre (**C**,**D**) and TownSuburbs (**E**,**F**) were acquired in a moderately structured environment of a small town (**B**). The trajectories estimated by PlaneLOAM are closer to the ground truth than their counterparts obtained from the open-source LOAM (**C**,**E**), while the ATE values for PlaneLOAM trajectories are much smaller (**D**,**F**).

**Figure 20 sensors-21-03445-f020:**
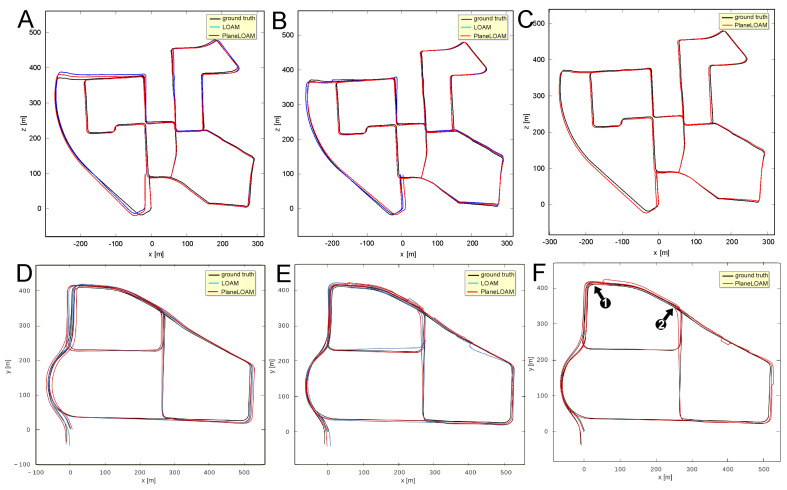
Estimated trajectories of the KITTI 00 (**A**–**C**) and MulRan DCC (**D**–**F**) sequences for the PlaneLOAM and open-source LOAM systems without any loop detection (**A**,**D**), with the SegMap loop detection method and two different approaches to loop closing: pose graph (**B**,**E**) or alignment of high-level features (**C**,**F**). Arrows point locations where the point clouds visualized in Figure 21 were acquired (see main text for details).

**Figure 21 sensors-21-03445-f021:**
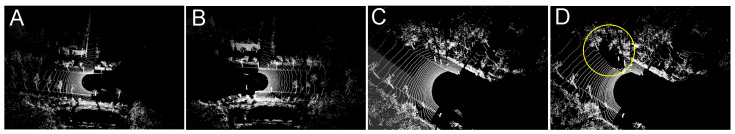
Visualization of example point clouds from the Mulran DCC sequence: a location visited four times while traversing a circular path with successful loop closures in spite of the reversed motion direction (**A**,**B**) and a location where the environment has changed due to a removed vehicle (encircled), but a loop closure was still detected (**C**,**D**).

**Figure 22 sensors-21-03445-f022:**
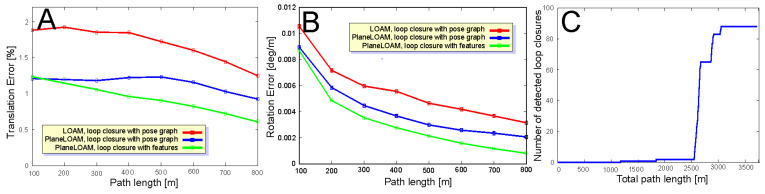
Influence of the loop closing method on the trajectory estimation accuracy for the KITTI 00 sequence shown according to the standard KITTI evaluation method [48] as the separate translation (**A**) and rotation (**B**) error. The number of detected loop closure proposals increases significantly while the vehicle progresses along the path and re-visits the already mapped areas (**C**).

**Table 1 sensors-21-03445-t001:** Comparison of ATE errors for publicly available LiDAR datasets recorded in urban environments.

Dataset	PlaneLOAM			LOAM (Open-Source)		
Sequence	${\mathrm{ATE}}_{\mathrm{RMS}}$	${\mathrm{ATE}}_{\max}$	$\sigma_{\mathrm{ATE}}$	${\mathrm{ATE}}_{\mathrm{RMS}}$	${\mathrm{ATE}}_{\max}$	$\sigma_{\mathrm{ATE}}$
KITTI 00	4.52 m	11.76 m	2.36 m	6.05 m	11.55 m	2.83 m
KITTI 05	3.21 m	8.04 m	1.59 m	3.39 m	11.26 m	1.86 m
KITTI 07	0.50 m	0.82 m	0.20 m	0.68 m	1.28 m	0.28 m
MulRan DCC	11.02 m	18.17 m	4.46 m	14.81 m	27.95 m	7.24 m

**Table 2 sensors-21-03445-t002:** Comparison of ATE errors for sequences recorded with a Sick MRS-6124 LiDAR.

Dataset	PlaneLOAM			LOAM (Open-Source)		
Sequence	${\mathrm{ATE}}_{\mathrm{RMS}}$	${\mathrm{ATE}}_{\max}$	$\sigma_{\mathrm{ATE}}$	${\mathrm{ATE}}_{\mathrm{RMS}}$	${\mathrm{ATE}}_{\max}$	$\sigma_{\mathrm{ATE}}$
TownCentre	4.93 m	14.05 m	2.10 m	6.05 m	24.90 m	6.48 m
TownSuburbs	4.91 m	8.22 m	1.79 m	8.71 m	18.37 m	3.04 m
Posnania	7.40 m	17.50 m	4.58 m	15.90 m	46.13 m	9.99 m

**Table 3 sensors-21-03445-t003:** Computation time necessary to process a single scan by the mapping thread.

Processing Step	Sub-Step	Computation Time [ms]			
		PlaneLOAM		LOAM (Open Source)	
Point clouds preparation		8.18		7.70	
Construction of kd-tree		2.49		75.80	
Point-to-line constraints		14.05		18.77	
	kd-tree search		8.45		10.87
	other computations		5.60		7.90
Point-to-plane constraints		13.11		18.30	
	kd-tree search		8.22		9.70
	other computations		4.89		8.60
Optimization time		0.45		0.57	
Point clouds post-processing		34.71		32.88	
Processing planar features		35.77		not applicable	
	adding points		22.76		
	updating features		2.22		
	deleting features		0.02		
	merging features		10.77		
Processing linear features		34.95		not applicable	
	adding points		15.53		
	updating features		16.65		
	deleting features		0.05		
	merging features		2.72		
Total time per scan		143.71		154.02	

**Table 4 sensors-21-03445-t004:** Comparison of ATE errors for different approaches to loop closing on the KITTI 00 sequence.

Approach to	PlaneLOAM			LOAM (Open Source)		
Loop Closing	${\mathrm{ATE}}_{\mathrm{RMS}}$	${\mathrm{ATE}}_{\max}$	$\sigma_{\mathrm{ATE}}$	${\mathrm{ATE}}_{\mathrm{RMS}}$	${\mathrm{ATE}}_{\max}$	$\sigma_{\mathrm{ATE}}$
none	4.52 m	11.76 m	2.36 m	6.05 m	11.55 m	2.83 m
pose graph	3.96 m	9.68 m	2.17 m	4.34 m	11.15 m	2.38 m
features alignment + BA	2.47 m	7.11 m	1.48 m	N/A	N/A	N/A

**Table 5 sensors-21-03445-t005:** Comparison of ATE errors for different approaches to loop closing on the MulRan DCC sequence.

Approach TO	PlaneLOAM			LOAM (Open-Source)		
Loop Closing	${\mathrm{ATE}}_{\mathrm{RMS}}$	${\mathrm{ATE}}_{\max}$	$\sigma_{\mathrm{ATE}}$	${\mathrm{ATE}}_{\mathrm{RMS}}$	${\mathrm{ATE}}_{\max}$	$\sigma_{\mathrm{ATE}}$
none	11.02 m	18.17 m	4.46 m	14.81 m	27.95 m	7.24 m
pose graph	7.41 m	19.23 m	3.59 m	9.77 m	27.51 m	4.26 m
features alignment + BA	5.40 m	13.78 m	2.01 m	N/A	N/A	N/A

## Data Availability

Not applicable.

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
