# Peer review of "Large-Scale LiDAR SLAM with Factor Graph Optimization on High-Level Geometric Features"

_sensors, 2021, doi:10.3390/s21103445_

Round 1

Reviewer 1 Report

The paper is well written and captures a real and interesting problem.

Some drawbacks must be accounted for before publication.

Although the main idea is interesting, there are situations that it seems not to work well. In fact, during the analysis of the mathematical framework and algorithms, it is possible to imagine that the approach is good when there are several closures and low dynamics (scene changes between the scannings). The results seem to corroborate this initial idea. 
To better understand the concept and its applicability, the authors should provide a closed circular path (rounding a circular plaza, for instance). 
Another necessary test is when trucks and/or busses are parked close to the looping closure venues.

Another critical issue is that no computational cost was evaluated. The authors should provide a point-by-point comparison with the original LOAM algorithm.

Reviewer 2 Report

My comments are as follows:

  1. This paper addresses the challenging map representation and loop closure problems for LiDAR SLAM systems in large urban environments. The proposed PlaneLOAM follows same software architecture of LiDAR Odometry and Mapping (LOAM) system, with additional loop closure and optimization module.
  2. The odometry thread is the same as LOAM. In mapping thread, PlaneLOAM proposes to group the planar and edge points of LOAM into high-level planar and linear features for map representation. In loop closure thread, a pose graph optimization pipeline with SegMap approach, is further improved by adding constraints from high-level feature matching.
  3. One main contribution of this work is that the proposed high-level geometric features allowLiDAR SLAM to solve loop closure using Bundle Adjustment optimization that can optimize sensor poses and feature parameters jointly.
  4. Experiments on KITTI, MulRan and author-collected datasets are performed to demonstrate that PlaneLOAM can achieve better trajectory accuracy compared to LOAM, with and without loop closure method. The comparison experiments and evaluation are well displayed and look convincing.
  5. In section 5.3 experiments (page 23), it is shown and discussed that for both PlaneLOAM and LOAM, proposed BA loop closing method can achieve better performance than pose-graph based method. Since mentioned also in section 2.1 (page 5) that there are already several pose-graph based loop-closing SLAM methods, direct comparison and quantitative evaluation withthose methods (including state-of-the-art method) could better demonstrate the capability of PlaneLOAM loop closure method.
  6. There are several strategies used for real-time performance, e.g. the kd-tree and point-to-feature matching in section 3.2.5. Nonetheless, it is still mentioned in the Conclusion (page 25) the increased processing time problem. Hence, some discussion or future efforts on this issue could be included.
  7. Minor error of annotation in Figure 6 B (page 12): d2min and d3min swapped. In Table2 (page 23), TownCenter line: value of ATEmax of LOAM not consistent with Figure 19D (page 22). Minor typo (page 24 line 668): ”Table. 4” instead of ”Fig. 4”.

  Overall the paper is well-written. The architecture of the PlaneLOAM system, algorithms and parameter selection are clearly and thoroughly described. The experiments evaluate the proposed SLAM method on KITTI, MulRan and author-collected datasets and have shown good improvement with respect to existed approach and applicability on large mixed indoor/outdoor environments.

Reviewer 3 Report

The manuscript entitled "Large-scale LiDAR SLAM with Factor Graph Optimization on High-Level Geometric Features" is based on a regular research work. Author proposed a new method that uses planar patches and line segments for map representation and employs factor graph optimization typical to state-of-the-art visual SLAM for the final map and trajectory optimization. The objective is well and the experiment were properly planned and measurements were appropriately executed.

The Table should use the three-line table and the caption of figure should not include the references.

One of the interesting questions is, how does that work on a road with a lot of people or a lot of cars? Whether will influence on the result? In addition, whether climate conditions affect the results? These contexts should be added in the discussion.

In addition, how to select the parameter values properly? Whether there is a method?

Round 2

Reviewer 1 Report

the paper can be published.